# Esophageal Lichen Planus—Contemporary Insights and Emerging Trends

**DOI:** 10.3390/biomedicines13112621

**Published:** 2025-10-26

**Authors:** Wolfgang Kreisel, Rebecca Diehl, Annegrit Decker, Adhara Lazaro, Franziska Schauer, Annette Schmitt-Graeff

**Affiliations:** 1Department of Medicine II, Gastroenterology, Hepatology, Endocrinology and Infectious Diseases, Faculty of Medicine, Medical Center, University of Freiburg, 79106 Freiburg, Germany; annegrit.decker@uniklinik-freiburg.de; 2Department of Dermatology, Faculty of Medicine, Medical Center, University of Freiburg, 79106 Freiburg, Germany; rebecca.diehl@uniklinik-freiburg.de (R.D.); franziska.schauer@uniklinik-freiburg.de (F.S.); 3Institute of Exercise and Occupational Medicine, Faculty of Medicine, Medical Center, University of Freiburg, 79106 Freiburg, Germany; adhara.lazaro@alumni.uni-heidelberg.de; 4Faculty of Medicine, Medical Center, University of Freiburg, 79106 Freiburg, Germany; annette.schmitt-graeff@uniklinik-freiburg.de

**Keywords:** autoimmunity, dysphagia, eosinophilic esophagitis, esophageal epidermoid metaplasia, esophagitis, JAK inhibitors, lichen planus, precancerous condition, squamous cell carcinoma

## Abstract

**Background**: Lichen planus (LP) is a common inflammatory disease affecting skin, mucous membranes, hairs, and nails, with an unpredictable course involving remissions and relapses. LP is a Type-I-Inflammation disease involving IFN-γ and IL-17 as key inflammatory mediators. **Materials and Methods:** We searched PubMed/MEDLINE and Google Scholar search engines for studies on the esophageal manifestation of lichen planus over an unlimited time frame. Articles were searched with combinations of Medical Subject Heading (MeSH) terms. Given the limited number of publications, no exclusion criteria were applied. **Results:** Esophageal lichen planus (ELP) is an underreported manifestation of LP that primarily affects middle-aged women. Its prevalence among LP patients remains to be defined. Though potentially clinically silent, ELP can significantly impact patient wellbeing and serve as a precursor to esophageal squamous cell carcinoma. While dysphagia is the primary symptom, the condition may also remain subclinical. The endoscopic hallmarks of ELP are mucosal denudation and tearing, trachealization, and hyperkeratosis. Chronic disease progression may lead to scarring esophageal stenosis. Histologically, ELP shows mucosal detachment, T-lymphocytic infiltrations, epithelial cell apoptosis (Civatte bodies), dyskeratosis, and hyperkeratosis. Fibrinogen deposits along the basement membrane zone distinguish ELP from various immunological esophageal diseases. There is currently no standardized therapy available. Topical steroids lead to symptomatic and histologic improvements in two-thirds of patients. Severe or refractory cases require immunosuppressive therapy, whereas JAK-inhibitors represent a promising emerging option. Endoscopic dilation helps symptomatic stenosis. Considering ELP’s precancerous potential, timely diagnosis and treatment are crucial in preventing complications, such as stenosis or invasive esophageal squamous cell carcinoma. **Conclusions:** ELP is an underdiagnosed and underreported manifestation of LP. While it may remain clinically silent, it can nevertheless significantly affect patients’ wellbeing and life expectancy. This narrative review aims to initiate multidisciplinary cooperation among gastroenterologists, dermatologists, oral health professionals, and histopathologists to support clinical diagnosis and management.

## 1. Lichen Planus

Lichen planus is a common inflammatory and autoimmune disease of the skin, mucous membranes, hair, and nails [1,2,3,4,5]. Its population-based prevalence was estimated to reach an average of 1.3% [1,6,7]. A meta-analysis showed varying prevalence of oral LP (0.57% in Asia, 1.39% in South America, and 1.68% in Europe) [6,7]. Lesions of skin, oral, and genital mucosa are the most frequent manifestations. The involvement of nails, scalp, genito-anal mucosa, eyes, ears, urinary bladder, or nasal mucosa can also be seen. Generally, the clinical course is benign, with fluctuating symptoms. Classic cutaneous LP presents as flat, reddish, itching papules on the face, arms, and wrists, with a tendency to progress to hyperpigmentation. Cutaneous LP is generally self-limiting and occasionally pruritic, with the majority of cases resolving spontaneously over months to years. Oral LP is considered the most predominant mucosal manifestation, affecting two-thirds of patients with cutaneous LP [8,9,10,11]. Oral LP differs markedly from its cutaneous counterpart in clinical behavior. It is typically chronic, treatment-resistant, and associated with considerable morbidity due to both its malignant potential and impact on quality of life. It exhibits characteristic fine white buccal lines (Wickham striae) or visible erosions or ulcerations on buccal mucosa, gingiva, palate, or tongue. Patients with oral LP generally complain of oral discomfort or pain. Importantly, oral LP is recognized as an oral potentially malignant disorder (OPMD), with reported rates of malignant transformation ranging from 0.44% to 2.28%. The risk is particularly elevated in patients with erosive or atrophic subtypes, tongue involvement, significant alcohol or tobacco consumption, or concurrent hepatitis C virus infection. Emerging research further suggests an association between vitamin D deficiency and oral LP development, highlighting potential nutritional factors in its pathogenesis [12].

Genital LP, on the other hand, may cause itching lesions on genital mucosa and may present with inflammation with erythema, erosions, plaque formation, and scarring. LP pemphigoides is a rare variant of LP, exhibiting characteristics of bullous pemphigoid with autoantibody deposition along the dermal–epidermal junction zone (e.g., reactivity against collagen XVII) [13]. Due to multiple organ manifestations, LP requires a multidisciplinary approach involving dermatologists, oral health professionals, gynecologists, gastroenterologists, and histopathologists [8,9,10,11]. The European guidelines for LP therapy have recently been published [14,15]. Timely diagnosis and therapy are essential as mucosal LP carries a recognized risk of malignant transformation.

## 2. Pathogenesis of LP

The exact pathogenesis of LP is not completely understood [16,17]. LP may be triggered by several drugs, e.g., NSAIDs, beta-blockers, ACE-inhibitors, and checkpoint inhibitors [18]. Agents such as amalgam, mercury, or gold may also induce oral LP [19,20]. Distinguishing true LP from lichenoid reactions is essential, as various drugs and dental restorative materials can induce lesions that mimic LP clinically and histologically. Lichenoid drug reactions typically resolve upon discontinuation of the offending agent, whereas true LP persists independently of external triggers. Additionally, oral LP may exhibit the Koebner phenomenon, in which chronic trauma or mechanical irritation can result in exacerbation or new lesion formation at sites of injury. Concomitant diabetes mellitus or smoking may influence clinical severity in LP [21]. An association with different autoimmune disorders, such as primary biliary cholangitis, autoimmune thyroiditis, myasthenia gravis, alopecia areata, vitiligo, thymoma, and autoimmune polyendocrinopathy [10,22,23,24], has been described. The assumed association with chronic hepatitis C remains controversial [25,26]. Psychological components may influence disease progression [27,28,29].

According to Eyerich et al., the immune reaction in immune-mediated skin diseases may be classified into six patterns [30]. LP is categorized as Type-I-Inflammation disease [30] driven by CD8 cytotoxic T-cells targeting the basal layer of keratinocytes, resulting in apoptosis and leading to the characteristic interface dermatitis [31,32,33,34]. In oral LP, certain key proteins are implicated in pathogenesis. Osteopontin, a multifunctional phosphoprotein, promotes T-cell activation and migration, thereby contributing to chronic inflammatory state. CD44, a cell surface adhesion molecule, facilitates lymphocyte homing and retention within the oral mucosa. Survivin, an anti-apoptotic protein, is dysregulated in oral LP and may contribute to the persistence of inflammatory infiltrates and altered keratinocyte survival. The central pathogenetic mechanism involves a sequential cascade. IFN-γ binds to its receptors on both keratinocytes and infiltrating lymphocytes. This binding activates JAK1 and JAK2 signaling in both cell types [35,36,37]. STAT1 is then activated, primarily in keratinocytes, leading to the upregulation of MHC-I expression. In active LP, JAK2 activation in lymphocytes and STAT-1 activation in keratinocytes may occur [38,39]. This process creates inflammatory amplification by enhancing keratinocyte’s responsiveness to inflammatory signals. The upregulation of IFN-γ, TNF-α, and IL-1, -6, and -8 causes basement membrane disruption [31,32,33,34]. The lichenoid infiltrates are predominantly composed of type-1 lymphocytes (Tc1-cells, Th1 cells, ILC1-cells, NKT-cells, and NK-cells), which react to exogenous or altered host-antigens presented by APCs, DCs, or keratinocytes. The inflammatory Th1-response leads to the secretion of proinflammatory cytokines (e.g., IL-23, IL-17, and IFN-γ), and the cytotoxic molecules perforin, granulysin, and granzyme B. This inflammatory cascade ultimately causes basement membrane disruption and keratinocyte apoptosis, resulting in distinct clinical manifestations depending on the anatomical location affected. In cutaneous LP, this process manifests as the characteristic violaceous, polygonal papules. When mucosal surfaces are involved, keratinocyte apoptosis results in painful erosive lesions that can significantly impact quality of life. In follicular lichen planus, damage to basal keratinocytes within hair follicles leads to irreversible scarring alopecia due to the destruction of follicular stem cells.

At present, there are no specific data on the pathogenesis of ELP. Since at least the histopathologic lesions of oral LP and ELP are comparable, a similar pathogenesis may be anticipated.

Accordingly, the use of JAK inhibitors in LP, as well as in other dermatological conditions with similar pathogenesis (e.g., vitiligo), provides a rational basis for targeted therapy compared with conventionally prescribed glucocorticosteroids or broader immunosuppressants [4,40,41,42].

## 3. Esophageal Lichen Planus

ELP was first described in 1982 [43]. It was initially regarded as a rare manifestation of LP [44,45,46]. However, further studies showed an esophageal involvement in up to 50% of patients with cutaneous or oral LP [47,48]. Despite growing interest in ELP, the number of cases in these studies were limited and patients were partly preselected and non-randomized in LP cohorts. ELP does not necessarily correlate with oral involvement [49]. However, oral LP is observed in most patients with severe ELP [50]. Esophageal manifestation is also associated with the occurrence of other mucosal involvement, such as genital LP [51]. The median age at presentation is 60 years and 70–80% of patients are female [49,52,53,54]. The clinical spectrum ranges from asymptomatic presentations to severe symptoms, such as dysphagia and bolus obstruction and upper gastrointestinal bleeding, and potentially ends in the development of esophageal squamous cell carcinoma. The percentage of ELP patients presenting with symptoms remains unknown. Previous studies found that 17–50% of patients with mild ELP did not report dysphagia [49,50]. However, treatment decisions should not rely exclusively on symptom severity, as asymptomatic ELP cases may also warrant therapy. Patients with symptomatic dysphagia should always undergo endoscopic evaluation.

Determining the true prevalence of ELP remains a challenge. This would require endoscopic screening in a large group of LP patients, regardless of disease localization or clinical symptoms. Consequently, even asymptomatic LP patients should undergo EGD. However, cost–benefit and risk–benefit considerations must be taken into account, even if screening would be of practical rather than only academic interest. Limiting assessment to LP patients with esophageal symptoms, e.g., dysphagia, would underestimate the true prevalence of ELP [55]. On the other hand, the selection bias of several studies could overestimate its population-based prevalence. Assuming that only 10% of all LP patients have esophageal involvement, the estimated prevalence in the general population could reach 0.1%, which would surpass the reported prevalence of eosinophilic esophagitis (0.04–0.05% in Western countries) [56].

Many aspects of ELP are still poorly understood and the disease itself remains underreported and underdiagnosed. Within the last two decades, ELP has gained increasing recognition among dermatologists, gastroenterologists, and histopathologists. Table 1 presents a representative selection of newer publications (only studies with more than 10 ELP patients and reviews) addressing various aspects of this disease.

## 4. Diagnostic Features of ELP

### 4.1. Clinical Symptoms

Dysphagia is the leading symptom in the majority of patients with ELP. It may include persistent swallowing difficulties, recurrent food impaction, and retrosternal chest pain [55]. Other reported complaints include odynophagia, heartburn, regurgitation, hoarseness, chronic unproductive cough, and weight loss. A substantial subset of ELP patients (between 20% [53] and more than 50%) does not present with any esophageal symptoms. As documented in a previous study, 94% of patients with endoscopically assessed severe ELP reported dysphagia, while only 44.4% of patients with mild ELP complained about dysphagia [49]. Conversely, LP patients without esophageal involvement may also have dysphagia, especially in LP with oral manifestation. Clinical differentiation between oropharyngeal and esophageal dysphagia could prove difficult; thus, endoscopic evaluation is warranted. Since, in a considerable percentage of ELP patients, the macroscopic aspect of the esophagus is ambiguous, histologic evaluation is necessary to confirm ELP [58]. ELP should be regarded as a differential diagnosis in all patients with dysphagia or food impaction.

### 4.2. Macroscopy

Denudation or sloughing of the esophageal mucosa was the endoscopic hallmark of ELP in nearly all studies. This may occur spontaneously or arise during the endoscopic procedure. “Trachealization” (endoscopically observed as ring formation, common in EoE) and the presence of a rough and whitish surface on the mucosa are nonspecific signs. The latter is the macroscopic correlate of a hyperkeratosis seen in histology [49,50,52,58]. As in other inflammatory esophageal diseases, chronic and uncontrolled inflammation can lead to stenoses or strictures. Figure 1 shows endoscopic images of mucosal alterations in ELP. These alterations were observed mainly in the middle third of the esophagus. As reflux esophagitis often occurs simultaneously with ELP, macroscopic and histologic alterations directly above the gastroesophageal junction may be ambiguous. Hence, biopsies should be taken at least 5 cm above the gastroesophageal junction. It is recommended that biopsies are obtained from all three thirds of the esophagus, regardless of macroscopic findings.

### 4.3. Key Morphologic and Immunophenotypic Features of ELP

Early histologic signs of esophageal involvement in LP may be subtle (Figure 2A–D). Predominantly CD3-positive T-lymphocytes aggregate at the interface between the superficial tunica propria and the squamous epithelium and spill over through the basal membrane into the basal epithelial layer (Figure 2C) [49,52,54]. The infiltration pattern is generally more pronounced in stromal papillae (Figure 2A). Deposits of fibrinogen, immunoglobulins, and/or complement may be detected by immunofluorescence studies [52]. The migration of lymphocytes in the epithelium is associated with intraepithelial edema, spongiosis, and apoptosis of scattered epithelial cells, resulting in “Civatte bodies” (Figure 2B). Intraepithelial splitting may be observed. In more advanced stages, dense subepithelial and intraepithelial lymphocytes blur the interface zone (Figure 2E,F). According to our observation, subepithelial T-cells predominantly express CD4 (Figure 2G), while intraepithelial T-cells may comprise various proportions of CD4- and CD8-positive cells (Figure 2D,H and Figure 3E). In the lamina propria, not only T-lymphocytes but also scattered B-cells, plasma cells, and macrophages may be present. The infiltrate may involve the higher epithelial layers and may be admixed with a small granulocytic component. Destruction of the basement membrane and partial or complete detachment of the squamous epithelium from the lamina propria may result in epithelial membranes that are, in contrast to sloughing esophagitis [61], not necrotic (Figure 2E and Figure 3A,B). Loss of the epithelium may trigger a more pronounced mixed-type inflammatory infiltrate of the lamina propria and the submucosa, followed by fibrosis and scarring. Adjacent to zones of denudation, the regenerative epithelium may show an increased Ki67-positive proliferation fraction (Figure 3F).

The squamous epithelium may undergo epidermoid metaplasia with the development of a granular layer and a cap of orthokeratosis or parakeratosis (see Figure 3 and Ref. [54]) Squamous dysplasia in hyperplastic or atrophic epithelium may be observed, involving the lower or the upper half of the epithelium, thus corresponding to low-grade or high-grade dysplasia/intraepithelial neoplasia. Such rare cases progress to an invasive squamous carcinoma.

### 4.4. ELP Histopathology: GENERAL Points and Caution

Histopathologists should be aware of the spectrum of diseases which can initiate a T-cell-rich inflammatory tissue injury of the esophageal mucosa that may be suggestive of ELP. Generally, ELP fits in the broad category of disorders for which the term lymphocyte-rich esophagitis has been coined [62]. Following an international expert consensus, this category includes three different morphologic forms, i.e., 1. lymphocytic, 2. lichenoid, and 3. lymphocyte-predominant esophagitis. According to this proposal for a standardized terminology, lymphocytic esophagitis (type 1) is defined by an intraepithelial infiltrate of ≥40 or, as previously stated, >20 lymphocytes per high power field (HPF) without a relevant neutrophilic component and without apoptotic epithelial cells (“Civatte bodies”) [62,63,64]. Figure 4A–C show an example of lymphocytic gastritis associated with lymphocytic esophagitis (Figure 4A,B) and numerous CD3-positive intraepithelial T-cells in a patient with Helicobacter pylori infection. The inflammatory infiltrate of the lymphocyte-predominant form (type 3, Figure 4D–F) is characterized by a mixed inflammatory infiltrate [65]. Examples include an EoE harboring abundant eosinophils (≥15 per HPF, Figure 4D) or an erosive Candida esophagitis rich in neutrophils and CD3-positive T-cells (Figure 4E,F). The lichenoid form of lymphocyte-rich esophagitis (type 2) encompasses both the manifestation of an ELP and pathogenetically different lesions with features suggestive of ELP [62,66,67]. Examples include viral infections such as those associated with HIV [68]. A common morphologic finding observed in both disorders is a band-like lymphocyte-rich infiltrate of the lamina propria involving the overlaying epithelium causing apoptosis of keratinocytes (“Civatte bodies”). However, the differentiation of a true ELP versus a lichenoid inflammation is of the utmost importance as these conditions have different prognoses and require different therapeutic approaches, and since mucosal LP is a precancerous condition with a documented risk of malignant progression [69,70,71]. It has been proposed that negative immunofluorescence studies may help discriminate true ELP from a lichenoid lesion, but these are infrequently performed [10,72,73]. Our findings indicate that complete detachment of the squamous epithelium from the tunica propria is rarely observed in non-ELP lichenoid esophagitis. Consideration of histopathologic and immunophenotypic features in light of clinical and endoscopic findings, as well as the pathophysiologic aspects, is necessary and could provide important adjuncts to the final diagnosis. This is particularly essential when esophageal biopsies show histologic features compatible with either ELP- or non-ELP-associated lichenoid inflammation. In the future, a more detailed understanding of the characteristic inflammatory cell populations, cytokines in the microenvironment, and molecular genetic epithelial alterations may contribute to the reliable differentiation between true ELP and its clinical mimickers.

### 4.5. Direct Immunofluorescence

Another important criterion in ELP constitutes the fibrinogen deposits along the basal membrane found in DIF. In oral diseases, linear fibrinogen deposition (or granular IgG and IgM deposits) in DIF could discriminate LP from other lichenoid lesions [73] and mucous membrane pemphigoid [10,72]. Therefore, positive results in the DIF support the conventional histopathologic diagnosis of ELP and, in turn, differentiate ELP from diseases like mucous membrane pemphigoid or pemphigus vulgaris in erosive stages. Furthermore, in cases with lichenoid infiltration of the esophagus without positive DIF and without extraesophageal manifestation of LP, the term lichenoid esophageal pattern may be preferred [67].

### 4.6. Proposed Criteria for Diagnosis of ELP

Comparable macroscopic and histologic criteria for the diagnosis of ELP have been well-documented in the literature. While some findings are typical of ELP, others show similarities with different esophageal disorders, such as eosinophilic esophagitis, lymphocytic esophagitis, and sloughing esophagitis [63,65,74,75,76,77,78,79,80,81]. We recently integrated the existing criteria and our findings into a comprehensive and reproducible scoring system (Table 2, taken from Ref. [54]; copyright with the authors). This scoring system makes the grading of esophageal lesions in LP patients (into no ELP, mild ELP, and severe ELP) feasible [53]. Simplified diagnostic criteria were suggested: (1) an endoscopic denudation of any grade, accompanied by either a positive histopathological finding or positive DIF for fibrinogen depositions; (2) at least one of the possible endoscopic signs (stenosis, hyperkeratosis, or trachealization), along with either a positive histopathological result or positive DIF result, or along with dermatological confirmation of LP manifestation in other areas and histological exclusion of other common esophageal differential diagnoses [55]. In 77 of 132 patients in the study by Ravi et al., the diagnosis of ELP was made based on clinical symptoms alone, without characteristic esophageal biopsies [34]. However, we advocate for the use of a semiquantitative grading system which incorporates characteristic histological features [53].

## 5. Differential Diagnoses

Among the inflammatory esophageal disorders, reflux esophagitis is the most commonly encountered [82]. However, clinicians must consider a wide range of differential diagnoses [61,62,74,83,84]. Infectious agents include Candida spp. and various viruses, e.g., HSV or CMV. Viral esophagitis is primarily associated with compromised immune function [85]. Chemical esophagitis may result from accidental ingestion of corrosive substances or drugs, while radiation-induced esophagitis is a therapy-related injury. Immunologically mediated esophageal disorders include Crohn’s disease [86], Behçet’s disease [87], graft-versus-host disease following allogeneic stem cell transplantation [88], immune checkpoint inhibitor-related esophagitis [89], and the well-characterized EoE [75,76,77,90]. Lymphocytic esophagitis (LE) is a rare form of immune-mediated esophagitis [62,65,78], first described in 2006. In contrast to ELP, LE is characterized by lymphocytic infiltration predominantly in the peripapillary regions, rather than the band-like distribution typical of ELP. Additionally, Civatte bodies, commonly seen in ELP, are absent in LE. In cases where a lichenoid infiltrate is observed, but DIF is negative and there are no cutaneous LP manifestations, the term lichenoid esophagitis or lichenoid esophageal pattern (LEP) may be applied [67]. Hussein et al. have questioned whether LE represents a distinct disease entity, proposing instead that it may reflect a histologic variant of more common esophageal conditions [91]. Notably, the endoscopic appearance of LE can closely resembles that of EoE. International consensus is needed to establish a standardized histopathological definition, develop a validated endoscopic severity scoring system, and define an evidence-based management algorithm. Sloughing esophagitis [92,93] represents another poorly defined subtype of esophagitis, characterized histologically by the presence of granulocytes and bacterial colonization [61]. Differential diagnoses should also include esophageal involvement in autoimmune cutaneous bullous diseases, such as mucous membrane pemphigoid (MMP) or pemphigus vulgaris (PV) [72]. Additionally, epidermolysis bullosa acquisita may present with an esophageal manifestation [94,95]. This review specifically focuses on ELP as an additional form of esophagitis with autoimmune features. Table 3 presents the differential diagnoses of inflammatory diseases of the esophagus.

## 6. Therapy

Therapeutic guidelines are established for cutaneous and oral LP [4,14,42]. However, large phase III studies are still lacking for this indication. Due to the scarcity of studies on ELP, no standardized or generally accepted therapeutic guidelines exist for this manifestation. Oral retinoids such as acitretin, although part of the standard therapeutic repertoire, have not been effective in preventing or treating ELP [49,73,96,97], with the exception of few cases using alitretinoin [98]. Understanding the pathogenesis of ELP could inform therapeutic decisions. Corticosteroids remain the cornerstone of treatment due to their broad suppression of the inflammatory cascade. However, targeted therapies that address specific pathogenic pathways may offer greater therapeutic precision and efficacy.

Topical corticosteroids constitute the standard treatment for ELP. Notably, fluticasone and budesonide have demonstrated promising clinical and/or endoscopic response rates of up to 75% [33,49,52,57]. Viscous syrups or gels may enhance mucosal adherence and improve response rates [49]. Orodispersible tablets developed for EoE might offer an alternative delivery option [99,100], although these are not currently approved for use in ELP. The intralesional injection of glucocorticosteroids, especially in cases involving inflammatory stenoses, has been effective in some cases [44,96,101]. Systemic corticosteroids can be considered for induction therapy in severe cases [102]. However, long-term therapy is limited by the risk of systemic side effects. In more severe or refractory cases that are unresponsive to topical corticosteroids, systemic immunosuppression may be required. Agents such as adalimumab, hydroxychloroquine, mycophenolate, azathioprine, cyclosporine, tacrolimus, and rituximab have all been reported, with varying degrees of success [40,96,103,104,105,106,107,108]. Nonetheless, some cases remain refractory to multiple lines of immunosuppression [109]. In one of our patients, multiple immunosuppressors failed to achieve sustained improvement. Cyclophosphamide provided a partial clinical benefit, while the JAK inhibitor tofacitinib ultimately led to stable clinical, endoscopic, and histologic remission [110]. Secukinumab, an anti-IL-17 antibody, showed efficacy in mucosal LP [111], suggesting potential utility in ELP, although further studies are needed to evaluate its role.

JAK inhibitors represent the most rational, pathogenesis-based therapeutic approach for LP by directly targeting the IFN-γ/JAK/STAT pathway central to disease pathogenesis [41,42,112,113,114]. This has been supported by several smaller studies and case series, and a recent review has summarized the outcomes of LP patients treated with a JAK inhibitor [112]. Emerging evidence from case reports also indicates a positive therapeutic response of ELP to JAK inhibition [32,110,115].

As ELP typically represents a manifestation of systemic or multilocular LP, treatment should be coordinated through a multidisciplinary approach, involving at least gastroenterologists, dermatologists, and histopathologists, to ensure optimal management and monitoring.

### 6.1. Complications: Esophageal Stenosis and Food Impaction

As in other inflammatory diseases of the gastrointestinal tract (e.g., reflux esophagitis, Crohn’s disease), chronic inflammation in ELP may lead to the development of inflammatory or scarring stenosis. In a few cases, anti-inflammatory therapy with budesonide alone has provided relief from inflammatory stenosis [49]. In cases of symptomatic scarring symptomatic stenosis of the esophagus, endoscopic dilation may be necessary [33,116]. When combined with anti-inflammatory treatment, dilation may be more effective, as it can reduce mucosal fragility, minimize mechanical stress, and help prevent recurrence of stenosis. Aby et al. reported that multiple dilations were required in most patients, with a low complication rate of 1.9% [58]. ELP should be considered as a potential cause of food impaction [117] or unexplained esophageal stenosis, particularly when other causes have been ruled out [118,119,120].

### 6.2. ELP as Precancerous Condition

LP is generally considered a benign condition with a low impact on life expectancy, although it can affect quality of life. However, mucosal LP is increasingly recognized as a precancerous condition, despite ongoing debate regarding the exact rate of malignant transformation [69,121,122,123,124]. The correlation between ELP and the development of esophageal squamous cell carcinoma (ESCC) is well documented. A growing number of case reports have described progression from inflammatory and hyperkeratotic lesions to squamous cell dysplasia/IEN or invasive ESCC [34,125,126]. Ravi et al. reported ESCC progression in 6 of 132 ELP patients during a 44-month follow-up period [34]. Notably, in our most recent prospective study, 10% of dysphagic ELP patients (2/21) developed ESCC [55].

ELP-associated esophageal precancerous squamous lesions are typically found in areas of “esophageal epidermoid metaplasia” (EEM) [127,128,129,130]. In low-grade dysplasia, cytologic and structural epithelial abnormalities are confined to the lower half of the esophageal epithelium, whereas high-grade dysplasia involves more than half of the epithelial cell layers and is characterized by lack of surface maturation. Therefore, endoscopically visible areas of EEM/leukoplakia should be systematically sampled for histologic evaluation, as these lesions are a hallmark of orthokeratotic dysplasia. Invasive ESCC may arise beneath or adjacent to EEM, emphasizing the need for careful endoscopic surveillance.

Molecular studies have further supported the neoplastic potential of ELP. According to Singhi et al. [128], TP53 mutations correlate with the presence of ESCC or progression of EEM to ESCC. In our cohort, p53 overexpression was frequently observed upon immunohistochemistry. However, additional molecular analyses are needed to identify reliable biomarkers for the early detection of high-risk ELP with the potential for malignant progression.

## 7. Proposal for Management of ELP

We recommend EGD in all patients with LP (whether presenting with cutaneous or mucosal manifestation) who report any esophageal symptoms. The diagnosis of ELP can be established using either the original criteria or the simplified diagnostic criteria (Table 2). Importantly, it must be emphasized that severe ELP can occur in asymptomatic individuals. Therefore, the threshold for performing EGD should be liberally defined in clinical practice. We recommend initial treatment of all ELP cases with topical corticosteroids and to reevaluate the response after approximately three months. In our clinical experience, a commonly used regimen consists of 0.5 mg budesonide in 5 mL viscous solution, administered three times a day, with gradual tapering based on clinical response. Decisions regarding further therapy should be guided by whether a clinical and/or histological remission has been achieved. If remission is not attained, systemic immunosuppressive therapy may be warranted. Systemic options include systemic glucocorticoids or traditional immunosuppressants such as azathioprine or mycophenolate. Emerging evidence also supports the use of JAK-inhibitors as a promising alternative in refractory cases. In patients with symptomatic esophageal stenosis unresponsive to anti-inflammatory therapy, endoscopic dilation is indicated. Esophageal candidiasis is a common adverse effect of steroid therapy and should be treated with antifungal agents. Endoscopic monitoring is recommended at least once annually, with frequency adjusted based on the individual clinical course. All patients diagnosed with ELP without known LP at other sites should undergo dermatological evaluation.

Currently, there is no consensus on the optimal management of asymptomatic ELP patients, particularly in cases presenting with esophageal hyperkeratosis, which may signal a risk for EEM, a potential precursor of ESCC. A “wait-and-see” approach may be appropriate in select cases [49,53]. However, in patients with confirmed EEM, we recommend EGD every six months to monitor for the development of dysplasia. In the event of dysplastic transformation, esophageal mucosal resection should be performed in a specialized center.

## 8. Emerging Trends

Multiple studies have shown positive outcomes with topically applied corticosteroids in the treatment of ELP. However, the optimal treatment duration and long-term maintenance strategies remain undefined. An orally dissolving budesonide preparation, approved for the treatment of EoE, represents a promising therapeutic option for ELP, though current evidence is limited to case reports [77,131].

Emerging data on the pathogenesis of LP suggest a disturbance in the IL12/23 cytokines and/or IL-17 axis in ELP, similar to mechanisms implicated in psoriasis or vitiligo [132,133,134,135,136]. Although IL-17 and IL-23 inhibitors have undergone phase II evaluation for (E)LP, these trials failed to achieve the primary endpoint for treatment response. In contrast, JAK inhibitors showed the most therapeutic potential, as they target the IFN-γ JAK/STAT pathway and have yielded superior results in LP therapy. Future therapy may be guided by treatment paradigms established in inflammatory bowel disease [137,138,139,140]. Potential candidates include ozanimod and etrasimod, S1P-receptor agonists recently approved for ulcerative colitis [141,142,143,144]. Another promising agent is deucravacitinib, a tyrosine-kinase 2-inhibitor that modulates IL-12 and IL-23 pathways and has demonstrated efficacy in other autoimmune diseases, such as Crohn’s disease, ulcerative colitis [145], and localized or systemic lupus erythematosus [146,147,148,149].

As with other immune-mediated diseases, environmental or lifestyle factors including psychological stress may play an important role. The identification of potential environmental factors (e.g., dental fillings with gold or amalgam) could offer new avenues for prevention and management.

Notably, the majority of ELP patients are postmenopausal women [49,55]. A correlation between menopause and oral LP is well established [150], possibly linked to declining estrogen and progesterone levels during perimenopause. Although this hormonal relationship has not been directly studied in ELP [151], it is known that postmenopausal women have a higher proinflammatory immune profile, rendering them more susceptible to inflammatory diseases [152].

While the progression from ELP to ESCC is increasingly reported, the reverse association has not been investigated. It would be valuable to assess the prevalence of undiagnosed ELP in ESCC patients, particularly in non-smokers, non-drinkers, and women, who lack classical ESCC risk factors. Such a study could further support the hypothesis that menopause and chronic mucosal inflammation may contribute to ESCC pathogenesis.

The oncogenic potential of inflammatory esophageal diseases warrants further exploration. Chronic reflux esophagitis can progress to adenocarcinoma of the esophago-gastric junction, typically via Barrett’s esophagitis, whereas chronic ELP may evolve into ESCC, often through a precursor stage of EEM/leukoplakia. In contrast, EoE, another immune-mediated esophagitis, is not considered a precancerous condition, highlighting the unique malignant potential of ELP among inflammatory esophageal diseases.

## Figures and Tables

**Figure 1 biomedicines-13-02621-f001:**
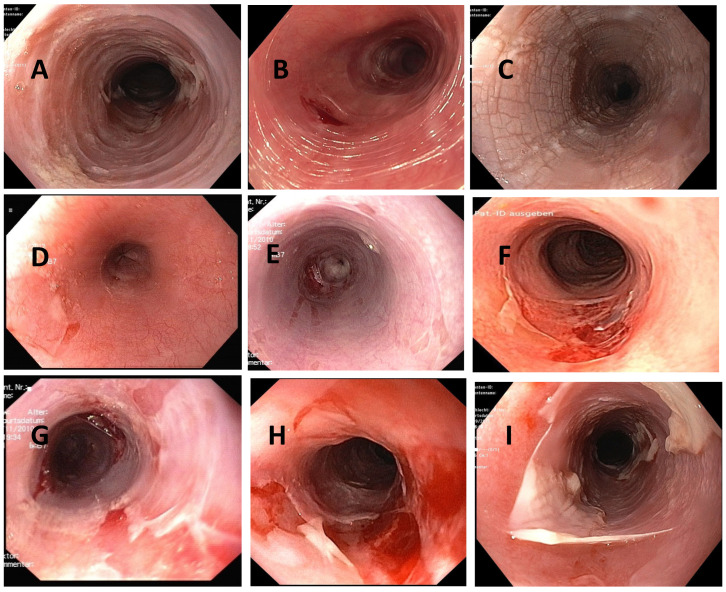
Endoscopic findings in esophageal lichen planus. (**A**): trachealization; (**B**): trachealization and fragile mucosa; (**C**): hyperkeratosis; (**D**,**E**): tearing; (**F**,**G**): tearing and localized denudation of the mucosa; (**H**,**I**): tearing and spacious denudation of the mucosa.

**Figure 2 biomedicines-13-02621-f002:**
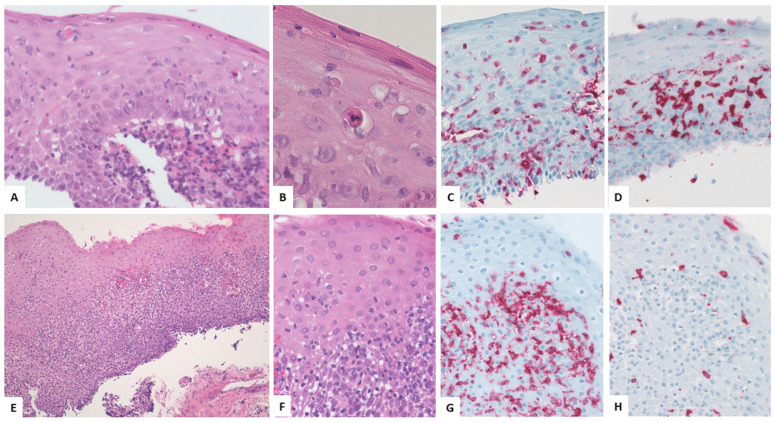
Histopathologic features of ELP: band-like lymphocytic infiltrate in the superficial tunica propria, most prominent at the rete ridges, spilling over to the squamous epithelium (**A**). Scattered epithelial apoptosis (Civatte bodies, (**B**)). Intraepithelial CD3-positive T-cells (**C**) enriched in a CD8-positive subpopulation (**D**). Focal detachment of the squamous epithelium from the tunica propria (**E**). Example of a case with intraepithelial lymphocytes (**F**) composed of CD4-positive T-cells (**G**) and only a small number of CD8-positive cells (**H**). (**A**,**B**,**E**,**F**): hematoxylin and eosin; (**C**,**D**,**G**,**H**): immunohistochemical stains for CD3, CD4, or CD8. Original magnification: (**A**,**F**) ×400, (**B**) ×1000, (**C**,**D**,**G**,**H**) ×250, (**E**) ×100.

**Figure 3 biomedicines-13-02621-f003:**
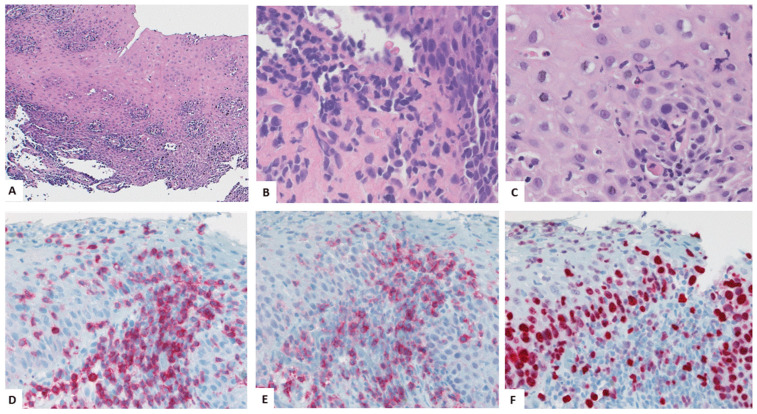
Dissection of the esophageal squamous epithelium at the junctional zone associated with confluent inflammatory infiltrates (**A**–**C**) Abundant CD3-positive T-cells (**D**) predominantly expressing CD4 in this case (**E**). Increased Ki67-positive proliferation fraction of the squamous epithelium (**F**). (**A**–**C**): hematoxylin and eosin; (**D**–**F**): immunohistochemical stains for CD3, CD4, or Ki67. Original magnification (**A**) ×100, (**B**,**C**) ×650, (**D**–**F**) ×250.

**Figure 4 biomedicines-13-02621-f004:**
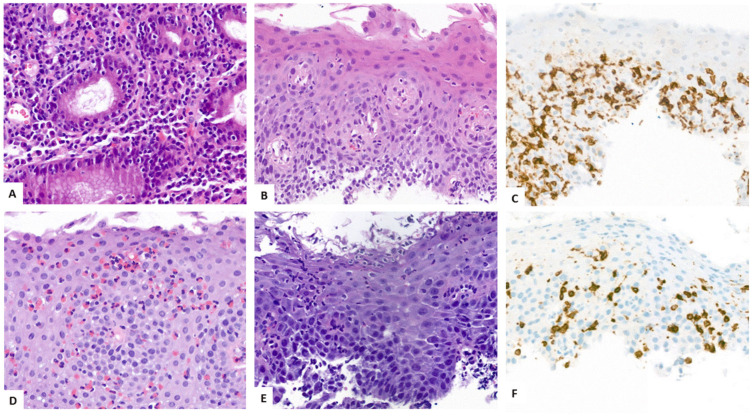
Conditions mimicking ELP: Lymphocytic gastritis (**A**) associated with a lichenoid esophagitis (**B**) with numerous CD3-positive T-cells (**C**); eosinophilic esophagitis (**D**) and erosive Candida esophagitis (**E**) characterized not only by a granulocytic but also by a CD3-positive T-cell component (**F**). (**A**,**B**,**D**): hematoxylin and eosin; (**E**): periodic acid Schiff reaction; (**C**,**F**): immunohistochemical stains for CD3. Original magnification: (**A**–**F**) ×400.

**Table 1 biomedicines-13-02621-t001:** Selection of important newer studies on esophageal lichen planus. (Only studies with more than ten ELP patients or reviews. Numbers in braces indicate number/percentage of patients from the cohort to which the criterion applies.)

Authors, References and Publication Year	Cohort/Study Design	Number of ELP Cases	Further Manifestation Sites of LP	Macroscopic Findings as Described in the Manuscript	Histologic Findings as Described in the Manuscript	Symptoms	Therapy
Quispel [48] 2009	24 LP patients	12	oral and/or cutaneous (all)	whitish papules (10) hyperemic lesions (3) mucosal detachment (2) submucosal plaques (3)	lymphohistiocytic infiltrations para-/hyperkeratosis hyperplasia Civatte bodies glycogen acanthosis	dysphagia (4) odynophagia (3) heart burn (3) regurgitation (2)	
Katzka [33] 2010	retrospective review (10 years) of data base/esophageal biopsies from patients with dysphagia	27 (female 92%)	oral (19) genital (13) cutaneous (3) ELP as initial manifestation (13)	strictures (18) proximal (11), distal (3), both (4), mucosal detachment (11) erythema, plaques, whitish mucosa, superficial ulcerations Koebner effect after dilation	lichenoid lymphocytic infiltration damage of epithelial basal layer Civatte bodies squamous cell carcinoma (1)	dysphagia (27) odynophagia (2)	dilatation of strictures (17) prednisone (6) intralesional corticosteroids (2) swallowed fluticasone/budesonide (2)
Fox [50] 2011	review of published ELP cases until 2009 (including 4 own cases)	72 (female 87%)	oral (89%) genital (42%) cutaneous (38%) scalp (7%) nails (3%) eyes (1%) ELP as initial manifestation (14)	pseudomembranes, bleeding, fragility, inflammation - proximal (64%) - distal (11%) - both (26%) stenosis (47%)	Lichenoid lymphocytic infiltrates dysplasia/squamous cell carcinoma (6%)	dysphagia (81%) odynophagia (24%) weight loss (14%) heart burn regurgitation hoarseness asymptomatic (17%)	
Podboy [57] 2017	retrospective analysis of a cohort of ELP-patients	40 (female 80%)	cutaneous (4) oral (19 genital (15) ELP as only manifestation (13)	strictures (29) ring formation (29) ulcerations (8) mucosal detachment (6) other mucosal lesions (14) squamous cell carcinoma (2)	esophagitis (20) focal ulcerations (13) mucosal hyperplasia (10) intraepithelial lymphocytic infiltrate (13), eosinophilia (13) dyskeratosis (11) DIF in 20 cases: positive, lichenoid (2) equivocal (5) not evaluable because of mucosal detachment (13)	dysphagia for solid food (32) even for fluids (8) odynophagia (6) reflux (1)	topical corticosteroids - budesonide in honey 2 × 3 mg (32) fluticasone spray 880 µg 2×/d (8) response rate: endoscopic (72.5%) clinical (62%)
Ravi [34] 2019	retrospective analysis of ELP patients	132 (female 80%)		“Clinical diagnosis” (77)	“Specific histology” (55) esophageal carcinoma (8)		response to topical steroids (84) immunosuppressive therapy (38)
Kern [52] 2016 Schauer [49] 2019	52 patients with proven LP on other site (♀ 75%)	34 - mild (18) - severe (16)	oral 78–100% in ELP 78% in non-ELP) genital 44–61% in ELP 6% in non-ELP cutaneous 25–44% in ELP 28% in non-ELP	mucosal detachment - iatrogenic (12) - spontaneous (16) hyperkeratosis (7) trachealization (10) stenosis/strictures (7)	epithelial detachment lymphocytic infiltration Civatte bodies dyskeratosis DIF: fibrinogen deposits (17) (85% in severe ELP)	dysphagia - severe ELP: 15 - mild ELP: 8	topical steroids (12) - budesonide gel 3 × 0.5 mg - fluticasone Stenosis: - topical steroids - dilation
Aby [58] 2023	Descriptive multicenter report	78 ELP (female 86%)	Oral (14) Skin (6) Multisystemic (18)	Strictures (42) Denudation (39) Narrow caliber esophagus (21)	Not listed	Not mentioned	PPI alone Topical steroids Systemic steroids Intralesional steroids PPI + intralesional + topical steroids Immunosuppressors.
Diehl [55] 2025	Prospective analysis of LP with dysphagia	21 ELP (female 71%)	Oral (17) Genital (9) Nail (8), Hair (7) Skin (5), Anal (1), Eye (1)	Denudation (13) Hyperkeratosis (9) Trachealization (15) Stenosis (13)	Civatte bodies (7) Dyskeratosis (12) Epithelial detachment (7) Lymphocytic infiltrate (16) DIF: fibrinogen deposits (13)	Dysphagia Food bolus obstruction Heart burn	Not listed in detail
Decker [54] 2022	Review						
Jacobs [31] 2022	Review						
Blonski [59] 2023	Review						
Ghai [60] 2025	Review						

**Table 2 biomedicines-13-02621-t002:** Diagnostic and grading for ELP, modified according to [49].

**Macroscopic-endoscopic criteria**	
**Specific signs** D Denudation/sloughing of the mucosa D1 Iatrogenic denudation (caused by biopsies) D2 Spontaneous localized denudation < 1 cm^2^ D3 Spontaneous spacious denudation > 1 cm^2^	**Possible signs** S Stenosis/stricture S1 Passable with standard endoscope S2 Not passable with standard endoscope H Hyperkeratosis (whitish, rough mucosa) T Trachealization N None of the criteria fulfilled
**Microscopic criteria—histopathology (HP) and direct immunofluorescence (F)**	
**HP** Sloughing of the epithelia (subepithelial, intraepithelial) Lymphocytic infiltrate, mainly T-lymphocytes, subepithelial, intraepithelial, junctional (region of the basal membrane) Intraepithelial apoptosis of keratinocytes “Civatte bodies” Dyskeratosis HP0 negative HP1 weakly positive HP2 positive HP3 strong positive	**F** Fibrinogen deposits along the basal membrane F0 no visible reaction F1 weak positive, discrete deposits visible F2 marked fibrinogen deposits along the basal membrane
**Grading**	
Severe LP ≥D2 and HP ≥ 1 OR ≥D2 and F ≥ 1 Mild ELP D1 and HP ≥ 1 and/or F ≥ 1 OR S, H, T, N and HP ≥ 1 and F ≥ 1 No ELP Criteria not fulfilled in a patient with LP on other localization	

**Table 3 biomedicines-13-02621-t003:** Differential diagnoses of ELP.

**Chemical or physical damages** Reflux esophagitis Chemical esophagitis (acids, leach) Radiation esophagitis Drug-induced esophagitis, e.g., NSAID, bisphosphonates, tetracyclines, KCl, ferric sulfate, ascorbic acid
**Infectious esophagitis** *Candida* spp. Viruses, e.g., Herpes simplex, CMV, HIV
**Immune mediated esophagitis** Eosinophilic esophagitis ELP Lymphocytic esophagitis Mucus membrane pemphigoid Pemphigus Lichenoid esophagitis Crohn’s disease GVHD Behçet’s disease Systemic sclerosis Immune checkpoint inhibitors
**Others** Epidermolysis bullosa congenita or acquisita Esophageal intramural pseudodiverticulosis (EIPD) Sloughing esophagitis

## Data Availability

No new data were created or analyzed in this study. Data sharing is not applicable to this article.

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
