# Peer review of "Esophageal Lichen Planus—Contemporary Insights and Emerging Trends"

_biomedicines, 2025, doi:10.3390/biomedicines13112621_

Round 1
Reviewer 1 Report (Previous Reviewer 2)
Comments and Suggestions for Authors
Dear Authors,
Thank you for the submission of the revised manuscript. The authors have incorporated the suggested revisions.
Best,
Author Response
Thank you very much for your positive comments
Reviewer 2 Report (New Reviewer)
Comments and Suggestions for Authors
General Comments
This is a comprehensive and well-structured narrative review on the important and underrecognized topic of Esophageal Lichen Planus (ELP). The authors successfully synthesize the current literature, covering pathogenesis, diagnosis, and management from a valuable multidisciplinary perspective. The manuscript is well-written and provides a clear overview for clinicians and researchers. The proposed diagnostic criteria and management algorithm are particularly strong contributions to the field.
Major Points
- The authors are encouraged to create systematic graphs (charts, bar charts, etc.) to provide visual and statistical support for the article.
- The manuscript presents an interdisciplinary approach that combines knowledge from gastroenterology, dermatology, and histopathology, all of which are essential for the treatment of the complex disease in question. However, there is a clear lack of generalizability in terms of treatment options, including drugs and drug delivery systems and new, highly effective formulations.
- A significant advantage of the proposed diagnostic and classification system (Table 2) is that it offers a practical basis for standardizing the diagnosis and evaluation of the severity of the condition, which is lacking at present. Would the authors be able to further refine this system, or do they believe it to be optimal from their perspective?
Inclusion of a clear clinical management plan (section 7) provides clinicians with real, evidence-based guidance, addressing a crucial gap in practice. Could the authors provide specific examples or statistical data from their work to support this aspect of their proposal? - The discussion of pathogenesis in this article is modern and correctly emphasizes the role of the IFN-γ/JAK/STAT signaling pathway. It provides strong arguments for the use of new treatments, such as JAK inhibitors. Please summarize the pathogenesis of this disease in the form of a diagram or flowchart.
Minor Points for Consideration:
- Authors are strongly advised to check the styles used according to the MDPI template, including fonts, colors, etc.
- The statement that ELP prevalence "could reach 0.1%" is an interesting projection but is based on a broad assumption. It may be beneficial to temper this statement, perhaps by rephrasing to more explicitly frame it as a hypothesis that underscores the need for future epidemiological studies.
- The "Emerging Trends" section is insightful. To further strengthen the discussion on novel therapies like S1P-receptor agonists, the authors could briefly expand on the specific mechanistic rationale for their potential application in ELP, linking them more directly to the known pathophysiology of lichen planus.
Author Response
Reviewer 2
Open Review
(x) I would not like to sign my review report
( ) I would like to sign my review report
Quality of English Language
( ) The English could be improved to more clearly express the research.
(x) The English is fine and does not require any improvement.
|
Is the work a significant contribution to the field? |
|
|
Is the work well organized and comprehensively described? |
|
|
Is the work scientifically sound and not misleading? |
|
|
Are there appropriate and adequate references to related and previous work? |
Comments and Suggestions for Authors
General Comments
This is a comprehensive and well-structured narrative review on the important and underrecognized topic of Esophageal Lichen Planus (ELP). The authors successfully synthesize the current literature, covering pathogenesis, diagnosis, and management from a valuable multidisciplinary perspective. The manuscript is well-written and provides a clear overview for clinicians and researchers. The proposed diagnostic criteria and management algorithm are particularly strong contributions to the field.
Major Points
- The authors are encouraged to create systematic graphs (charts, bar charts, etc.) to provide visual and statistical support for the article.
We thank the reviewer for this suggestion and recognize that additional visual elements could potentially enhance the manuscript’s presentation. However, at this stage—given the nature of the resubmission and the scope of revisions already completed—we have opted to retain the current format, which we believe adequately conveys the findings in a clear and interpretable manner. We hope the manuscript in its present form will be considered acceptable for publication.
- The manuscript presents an interdisciplinary approach that combines knowledge from gastroenterology, dermatology, and histopathology, all of which are essential for the treatment of the complex disease in question. However, there is a clear lack of generalizability in terms of treatment options, including drugs and drug delivery systems and new, highly effective formulations.
We appreciate the reviewer’s insightful observation regarding the current limitations in generalizing treatment options. As we emphasize throughout the manuscript—including in Table 1—the overall evidence base remains sparse. Fewer than 1,000 patients with ELP have been described in the literature to date, with wide variability in patient characterization, diagnostic criteria, and therapeutic approaches across centers. Given this heterogeneity, no standardized treatment recommendations can currently be made. Existing treatment strategies are largely center-specific and often influenced by local experience or preference, rather than evidence-based consensus. The goal of this review is to synthesize the available data, highlight these significant gaps, and support the development of a more structured framework for diagnosis and, eventually, treatment. Our proposed diagnostic criteria are intended as a step toward this aim and to encourage more systematic investigation moving forward.
- A significant advantage of the proposed diagnostic and classification system (Table 2) is that it offers a practical basis for standardizing the diagnosis and evaluation of the severity of the condition, which is lacking at present. Would the authors be able to further refine this system, or do they believe it to be optimal from their perspective?
We thank the reviewer for this encouraging assessment of our proposed diagnostic and classification system. Based on our experience, the criteria presented are well established, reproducible, and provide a solid foundation for both clinical application and future research. At this stage, we consider the system to be adequate and fit for purpose. Further refinement—particularly of histopathological descriptors—would likely require large-scale, multicenter validation and consensus, given the degree of expertise and interpretive nuance involved. We hope that the current proposal will serve as a useful starting point for such future efforts and are pleased that the reviewer finds it valuable.
Inclusion of a clear clinical management plan (section 7) provides clinicians with real, evidence-based guidance, addressing a crucial gap in practice. Could the authors provide specific examples or statistical data from their work to support this aspect of their proposal?
We thank the reviewer for highlighting the clinical value of Section 7. We fully agree that the inclusion of concrete data or case-based examples would further strengthen the management framework. Unfortunately, at present, the available literature does not provide sufficient consistency or statistical robustness to support detailed, evidence-based recommendations beyond what we have already summarized. Our aim with this review is to raise awareness of ELP as an underrecognized entity and to encourage the kind of systematic studies that would eventually allow for the type of data-driven guidance the reviewer rightly points to. Within the limits of the current evidence base, we believe the clinical recommendations presented represent the most comprehensive synthesis achievable at this time.
- The discussion of pathogenesis in this article is modern and correctly emphasizes the role of the IFN-γ/JAK/STAT signaling pathway. It provides strong arguments for the use of new treatments, such as JAK inhibitors. Please summarize the pathogenesis of this disease in the form of a diagram or flowchart.
We appreciate the reviewer’s recognition of our discussion on the pathogenesis of ELP. While much of the current understanding is inferred from studies on lichen planus (LP), a direct, evidence-based transfer of these mechanisms to ELP remains unproven. We are planning further investigations into the inflammatory mediators (e.g., interleukins) and the immune cells involved—primarily T lymphocytes—but the data are not yet available. Given this, we are cautious about presenting a flowchart at this stage, as it would be largely speculative and might imply a level of mechanistic certainty that the current evidence does not yet support. We hope the discussion as it stands provides a balanced and up-to-date overview while acknowledging the significant gaps that remain.
Minor Points for Consideration:
- Authors are strongly advised to check the styles used according to the MDPI template, including fonts, colors, etc.
Thank you for the reminder. We have already reviewed the manuscript carefully to ensure that it adheres to the MDPI formatting guidelines, including font styles, colors, and overall layout.
- The statement that ELP prevalence "could reach 0.1%" is an interesting projection but is based on a broad assumption. It may be beneficial to temper this statement, perhaps by rephrasing to more explicitly frame it as a hypothesis that underscores the need for future epidemiological studies.
Thank you for pointing that out. We have revised the relevant sentence to clearly frame the prevalence estimate as a hypothesis and to emphasize the need for well-designed epidemiological studies to better define the true prevalence of ELP. Compare lanes 157 – 159.
- The "Emerging Trends" section is insightful. To further strengthen the discussion on novel therapies like S1P-receptor agonists, the authors could briefly expand on the specific mechanistic rationale for their potential application in ELP, linking them more directly to the known pathophysiology of lichen planus.
We thank the reviewer for this thoughtful suggestion. As outlined in the pathogenesis section, the JAK/STAT signaling pathway is currently the most clearly implicated mechanism, which supports the potential use of JAK inhibitors. Regarding S1P receptor modulators, we do observe certain parallels with chronic inflammatory bowel diseases, where these agents have shown therapeutic benefit. However, the exact mechanistic rationale—particularly in the context of ELP or even LP—remains speculative at this point.
Submission Date
30 September 2025
Date of this review
15 Oct 2025 13:14:37
Formularende
© 1996-2025 MDPI (Basel, Switzerland) unless otherwise stated
This manuscript is a resubmission of an earlier submission. The following is a list of the peer review reports and author responses from that submission.
Round 1
Reviewer 1 Report
Comments and Suggestions for Authors
This is a nice review integrating dermatological, gastroenterological, and histopathological perspectives, which reflects the multidisciplinary approach necessary for managing this disease. The inclusion of extensive tables summarizing previous cohorts and histological findings is a strength. Some comments below
Major comments
The assertion that up to 50% of LP patients may have esophageal involvement is derived from highly selected cohorts and is likely an overestimate. This should be presented more cautiously.
The recommendation for endoscopic screening of all LP patients is not sufficiently justified by current evidence. A cost–benefit and risk–benefit discussion would be appropriate before advocating such a broad strategy.
The immunological mechanisms are well described but somewhat repetitive. The review would benefit from a clearer schematic summarizing the IFN-γ/JAK/STAT axis, the role of CD8+ T cells, and how this mechanistically links to both clinical features and therapeutic targets.
The section conflates cutaneous/oral LP data with oesophageal disease. It should be clarified which mechanisms are confirmed in ELP versus extrapolated.
The discussion of differential diagnoses is thorough, but the practical guidance on how to distinguish ELP from eosinophilic or lymphocytic oesophagitis could be more concise.
The therapeutic evidence is largely anecdotal or derived from case series. This limitation should be more explicitly acknowledged.
The review appropriately highlights the risk of progression to squamous cell carcinoma. However, the true incidence remains uncertain and the figures quoted (6–10%) derive from limited cohorts with possible surveillance bias.
Recommendations for surveillance intervals (e.g., annual EGD, 6-monthly for EEM) should be framed as expert opinion rather than evidence-based guidelines.
Minor Comments
Some sections are overly detailed (e.g., histopathology descriptions, cytokine pathways) and may overwhelm a general readership. Summarizing these with figures and focusing on the clinically relevant aspects would improve readability.
Occasional typographical errors and inconsistencies (e.g., “INF-y” instead of “IFN-γ”) should be corrected.
Table 1 is useful but could be condensed; many case details are not critical for a narrative review.
Author Response
To reviewer #1,
We thank the reviewer for their helpful comments. We have changed the manuscript according to their suggestions. Several paragraphs were completely new written or modified.
The style of the manuscript was better adapted to the style of the journal. Particularly, the references were written as prescribed for “Biomedicines”.
All changes are marked in red.
We think that the correction and modifications have considerably improved the manuscript
Open Review #1
( ) I would not like to sign my review report
(x) I would like to sign my review report
Quality of English Language
( ) The English could be improved to more clearly express the research.
(x) The English is fine and does not require any improvement.
|
Is the work a significant contribution to the field? |
|
|
Is the work well organized and comprehensively described? |
|
|
Is the work scientifically sound and not misleading? |
|
|
Are there appropriate and adequate references to related and previous work? |
Comments and Suggestions for Authors
This is a nice review integrating dermatological, gastroenterological, and histopathological perspectives, which reflects the multidisciplinary approach necessary for managing this disease. The inclusion of extensive tables summarizing previous cohorts and histological findings is a strength. Some comments below
Major comments
The assertion that up to 50% of LP patients may have esophageal involvement is derived from highly selected cohorts and is likely an overestimate. This should be presented more cautiously.
Answer: P3, Line 136 the wording was adapted to the following “……and patients were partly preselected and non-randomized…”
The recommendation for endoscopic screening of all LP patients is not sufficiently justified by current evidence. A cost–benefit and risk–benefit discussion would be appropriate before advocating such a broad strategy.
Answer: We do agree with this point and made the following changes. P4, Line 152. “However, cost-benefit and risk-benefit considerations must be taken into account, even if this would be of academic interest.
The immunological mechanisms are well described but somewhat repetitive. The review would benefit from a clearer schematic summarizing the IFN-γ/JAK/STAT axis, the role of CD8+ T cells, and how this mechanistically links to both clinical features and therapeutic targets.
Answer: P3, Line 81-131: The paragraph “Pathogenesis” was completely new written. We think, the role of the IFN-γ/JAK/STAT axis, the role of CD8+ T cells is clearer described.
The section conflates cutaneous/oral LP data with oesophageal disease. It should be clarified which mechanisms are confirmed in ELP versus extrapolated.
Answer: We inserted some remarks on this issue in the paragraph on pathogenesis. Lines 125 – 127.At present, there are no specific data on pathogenesis of ELP. Since at least histopathologic lesions of oral LP and ELP are comparable, a similar pathogenesis may be anticipated.
The discussion of differential diagnoses is thorough, but the practical guidance on how to distinguish ELP from eosinophilic or lymphocytic oesophagitis could be more concise.
Answer: Since these disorders may show considerable overlap, only the combination of endoscopic findings, a properly representative biopsy, expert histopathological assessment, and, if available, direct immunofluorescence allows for a more definitive diagnosis.
In paragraph 5 on differential diagnosis, lymphocytic esophagitis was described as follows: P. 8, line 335 and following “Lymphocytic esophagitis (LE) is a rare form of immune-mediated esophagitis [64,67,80] first described in 2006. In contrast to ELP, LE is characterized by lymphocytic infiltration predominantly in the peripapillary regions, rather than the band-like distribution typical of ELP. Additionally, Civatte bodies, commonly seen in ELP, are absent in LE. In cases where a lichenoid infiltrate is observed, but DIF is negative and there are no cutaneous LP manifestations, the term lichenoid esophagitis or lichenoid esophageal pattern (LEP) may be applied [69]. Hussein et al. have questioned whether LE represents a distinct disease entity, proposing instead that it may reflect a histologic variant of more common esophageal conditions [94]. Notably, the endoscopic appearance of LE can closely resembles that of EoE. International consensus is needed to establish a standardized histopathological definition, develop a validated endoscopic severity scoring system, and define an evidence-based management algorithm.“
The therapeutic evidence is largely anecdotal or derived from case series. This limitation should be more explicitly acknowledged.
Answer: Thank you for the comment. We addressed this on page 10, Line 358 and following. “Therapeutic guidelines are established for cutaneous and oral LP [4,14,42]. However, large phase III studies are still lacking for this indication. Due to the scarcity of studies on ELP, no standardized or generally accepted therapeutic guidelines exist for this manifestation.”
The review appropriately highlights the risk of progression to squamous cell carcinoma. However, the true incidence remains uncertain and the figures quoted (6–10%) derive from limited cohorts with possible surveillance bias.
Answer: this is correct. We added that in our prospective cohort, patient with dysphagia were preselected.
Recommendations for surveillance intervals (e.g., annual EGD, 6-monthly for EEM) should be framed as expert opinion rather than evidence-based guidelines.
Answer: The surveillance recommendations are aligned with those established for Barrett’s esophagus with and without dysplasia and were therefore set at the stated intervals. Naturally, further experience is needed to increase the level of evidence.
Minor Comments
Some sections are overly detailed (e.g., histopathology descriptions, cytokine pathways) and may overwhelm a general readership. Summarizing these with figures and focusing on the clinically relevant aspects would improve readability.
Answer: An adequate histopathologic and immunohistochemical evaluation of esophageal biopsies is an important tool to identify cases of ELP as opposed to other types of lymphocyte predominant esophagitis. The distinction impacts patient counselling and therapy and should be mandatory for patients enrolled in prospective studies and/or clinical trials. This current in-depth paragraph aims to increase the awareness of pathologists for the characteristic morphologic features of ELP thus helping to arrive at a correct diagnosis. Therefore, we would propose to preserve the submitted paragraph if the reviewers agree.
We cancelled Figure 6 (DIF). DIF is not regularly performed.
Occasional typographical errors and inconsistencies (e.g., “INF-y” instead of “IFN-γ”) should be corrected.
Answer: Thank you, we have corrected the typographical errors within the manuscript.
Table 1 is useful but could be condensed; many case details are not critical for a narrative review.
Answer: We think, the details presented in Table 1 are helpful for the user.
Reviewer 2 Report
Comments and Suggestions for Authors
Dear Authors,
Thank you for the submission of your manuscript. The manuscript describes about Esophageal lichen planus, an uncommon and frequently under diagnosed inflammatory disorder predominantly affecting postmenopausal women. It is characterized by mucosal changes such as denudation and trachealization, with diagnosis relying on endoscopic evaluation corroborated by histopathology. Topical corticosteroids and systemic immunosuppressants are the mainstay of therapy, although emerging therapies like JAK inhibitors have demonstrated promise in recalcitrant cases. Early diagnosis and vigilant follow-up are crucial to reduce the risk of progression to esophageal squamous cell carcinoma.
Please find my comments-
ABSTRACT
- Please provide a structured Abstarct encompassing Background, Materials and Methods, Results and Conclusion
- Keywords should be arranged alphabetically, and separated by commas
INTRODUCTION
- LP is a chronic inflammatory, autoimmune disorder (Not just autoimmune). Please change this.
- Briefly, differentiate between cutaneous and oral LP, and mention that OLP is regarded as an oral potentially malignant disorder (OPMD)
Add ‘Cutaneous LP tends to be self-limiting and only occasionally pruritic, whereas OLP is typically chronic, treatment-resistant, and associated with significant morbidity due to its malignant potential. OLP is recognized as an OPMD, with reported malignant transformation rates ranging from 0.44% to 2.28%. The risk is particularly elevated in erosive or atrophic subtypes, tongue involvement, individuals with high alcohol/tobacco use, and those with concurrent hepatitis C virus infection.
Cite https://pmc.ncbi.nlm.nih.gov/articles/PMC9687323/
PATHOGENESIS OF LP
- Mention the differentiating features of LP with lichenoid reactions (as various drugs and restorative materials are involved in the etio-pathogenesis).
- Mention koebners phenomena as chronic trauma may result in exacerbation of OLP lesions.
- Briefly, explain the role of Osteopontin, CD44, and Survivin proteins play a role in OLP pathogenesis.
- There is no methodology mentioned. Although, it is narrative review, the basic search strategy should be explicitly mentioned. Add a section on ‘MATERIALS AND METHODS’ including what search engines were used, time frame, keywords employed, inclusion and exclusion criteria etc.
- Tables should be better represented. Take care of the extra line spacing.
- Please modify Table 1 as Author(S), year; No. of ELP cases, M/F; other LP affected sites, followed by Signs and symptoms column.
- In the therapy section, table 1; please use a uniform pattern, either short forms (syst. Ster., ,
PPI, etc followed by the full form in the legend) or full forms.
REFERENCES
- References require MAJOR reframing. The journal guidelines for references should be followed.
- Remove PMIDs and [before doi]
- Doi/url to be placed after all the references
- Reference 19,21,23,24,26,47,50,51,57 etc are incomplete. Please check the complete list of references
- et al. To be used after the names of 10 authors (see the journal guidelines)
- Few of the cited references are too old. 1980, 84 etc. Please try replacing them with recent references.
Author Response
To reviewer #2,
We thank the reviewer for their helpful comments. We have changed the manuscript according to their suggestions. Several paragraphs were completely new written or modified.
The style of the manuscript was better adapted to the style of the journal. Particularly, the references were written as prescribed for “Biomedicines”.
All changes are marked in red.
We think that the corrections and modifications have considerably improved the manuscript.
Open Review #2
( ) I would not like to sign my review report
(x) I would like to sign my review report
Quality of English Language
( ) The English could be improved to more clearly express the research.
(x) The English is fine and does not require any improvement.
|
Is the work a significant contribution to the field? |
|
|
Is the work well organized and comprehensively described? |
|
|
Is the work scientifically sound and not misleading? |
|
|
Are there appropriate and adequate references to related and previous work? |
Comments and Suggestions for Authors
Dear Authors,
Thank you for the submission of your manuscript. The manuscript describes about Esophageal lichen planus, an uncommon and frequently under diagnosed inflammatory disorder predominantly affecting postmenopausal women. It is characterized by mucosal changes such as denudation and trachealization, with diagnosis relying on endoscopic evaluation corroborated by histopathology. Topical corticosteroids and systemic immunosuppressants are the mainstay of therapy, although emerging therapies like JAK inhibitors have demonstrated promise in recalcitrant cases. Early diagnosis and vigilant follow-up are crucial to reduce the risk of progression to esophageal squamous cell carcinoma.
Please find my comments-
ABSTRACT
- Please provide a structured Abstarct encompassing Background, Materials and Methods, Results and Conclusion
Answer: Thank you for the advice: the changes were made.
- Keywords should be arranged alphabetically, and separated by commas
Answer: Keywords were change to alphabetical order and separated by commas.
INTRODUCTION
- LP is a chronic inflammatory, autoimmune disorder (Not just autoimmune). Please change this.
Answer: We agree, inflammatory and autoimmune was added.
- Briefly, differentiate between cutaneous and oral LP, and mention that OLP is regarded as an oral potentially malignant disorder (OPMD)
Answer: Thank you. We adapted that point, which is highly relevant.
Add ‘Cutaneous LP tends to be self-limiting and only occasionally pruritic, whereas OLP is typically chronic, treatment-resistant, and associated with significant morbidity due to its malignant potential. OLP is recognized as an OPMD, with reported malignant transformation rates ranging from 0.44% to 2.28%. The risk is particularly elevated in erosive or atrophic subtypes, tongue involvement, individuals with high alcohol/tobacco use, and those with concurrent hepatitis C virus infection.
Cite https://pmc.ncbi.nlm.nih.gov/articles/PMC9687323/
Answer: thank you, we change the paragraph and added the reference.
PATHOGENESIS OF LP
- Mention the differentiating features of LP with lichenoid reactions (as various drugs and restorative materials are involved in the etio-pathogenesis).
- Mention koebners phenomena as chronic trauma may result in exacerbation of OLP lesions.
- Briefly, explain the role of Osteopontin, CD44, and Survivin proteins play a role in OLP pathogenesis.
Answer: The paragraph “Pathogenesis” was completely new written. All issues were addressed.
- There is no methodology mentioned. Although, it is narrative review, the basic search strategy should be explicitly mentioned. Add a section on ‘MATERIALS AND METHODS’ including what search engines were used, time frame, keywords employed, inclusion and exclusion criteria etc.
Answer: We inserted a remark in the Abstract
- Tables should be better represented. Take care of the extra line spacing.
- Please modify Table 1 as Author(S), year; No. of ELP cases, M/F; other LP affected sites, followed by Signs and symptoms column.
- In the therapy section, table 1; please use a uniform pattern, either short forms (syst. Ster., ,
Answer: We modified Table 1.
PPI, etc followed by the full form in the legend) or full forms.
REFERENCES
- References require MAJOR reframing. The journal guidelines for references should be followed.
- Remove PMIDs and [before doi]
- Doi/url to be placed after all the references
- Reference 19,21,23,24,26,47,50,51,57 etc are incomplete. Please check the complete list of referencesschauer
- et al. To be used after the names of 10 authors (see the journal guidelines)
- Few of the cited references are too old. 1980, 84 etc. Please try replacing them with recent references.
Answer: We adapted the style of reference to the style in “Biomedicines”.
Additionally, we found many inaccuracies and several duplicates. Ref. 42 from 1982 is the first description of ELP and should be mentioned.
Round 2
Reviewer 2 Report
Comments and Suggestions for Authors
Dear Authors,
Thank you for the submission of the revised manuscript incorporating the suggested changes.
Best wishes,